# Machine Learning to Discern Interactive Clusters of Risk Factors for Late Recurrence of Metastatic Breast Cancer

**DOI:** 10.3390/cancers14010253

**Published:** 2022-01-05

**Authors:** Juan Luis Gomez Marti, Adam Brufsky, Alan Wells, Xia Jiang

**Affiliations:** 1Department of Pathology, University of Pittsburgh, Pittsburgh, PA 15213, USA; jug68@pitt.edu; 2R&D Service, Pittsburgh VA Health System, Pittsburgh, PA 15240, USA; 3Department of Medicine, University of Pittsburgh, Pittsburgh, PA 15213, USA; brufskyam@upmc.edu; 4Hillman Cancer Center, University of Pittsburgh Medical Center, Pittsburgh, PA 15232, USA; 5Department of Biomedical Informatics, University of Pittsburgh, Pittsburgh, PA 15206, USA; xij6@pitt.edu

**Keywords:** metastatic breast cancer, metastasis, causal learning, machine learning, Markov Blanket and Interactive Risk Factor Learner (MBIL), risk factors

## Abstract

**Simple Summary:**

Breast cancer is the most frequently diagnosed cancer and second leading cause of cancer-related death among women worldwide. After initial tumor resection, breast cancer may recur locally and/or in distant organs within several months to years or even decades. Multiple methods exist to prognosticate disease progression in the early months and years after diagnosis. However, further efforts are needed to identify risk factors that relate to recurrence beyond the initial 5-year window. In this study, we applied machine learning to retrieve single and interactive clinical and pathological risk factors of 5-, 10- and 15-year metastases.

**Abstract:**

Background: Risk of metastatic recurrence of breast cancer after initial diagnosis and treatment depends on the presence of a number of risk factors. Although most univariate risk factors have been identified using classical methods, machine-learning methods are also being used to tease out non-obvious contributors to a patient’s individual risk of developing late distant metastasis. Bayesian-network algorithms can identify not only risk factors but also interactions among these risks, which consequently may increase the risk of developing metastatic breast cancer. We proposed to apply a previously developed machine-learning method to discern risk factors of 5-, 10- and 15-year metastases. Methods: We applied a previously validated algorithm named the Markov Blanket and Interactive Risk Factor Learner (MBIL) to the electronic health record (EHR)-based Lynn Sage Database (LSDB) from the Lynn Sage Comprehensive Breast Center at Northwestern Memorial Hospital. This algorithm provided an output of both single and interactive risk factors of 5-, 10-, and 15-year metastases from the LSDB. We individually examined and interpreted the clinical relevance of these interactions based on years to metastasis and reliance on interactivity between risk factors. Results: We found that, with lower alpha values (low interactivity score), the prevalence of variables with an independent influence on long-term metastasis was higher (i.e., HER2, TNEG). As the value of alpha increased to 480, stronger interactions were needed to define clusters of factors that increased the risk of metastasis (i.e., ER, smoking, race, alcohol usage). Conclusion: MBIL identified single and interacting risk factors of metastatic breast cancer, many of which were supported by clinical evidence. These results strongly recommend the development of further large data studies with different databases to validate the degree to which some of these variables impact metastatic breast cancer in the long term.

## 1. Introduction

Women who are diagnosed with invasive breast cancer will likely present with distant recurrence in the years after diagnosis [1]. Patterns and time to recurrence vary depending on tumor subtypes and the presence of concomitant biomarkers, as well as other clinical risk factors. In this regard, while recurrence occurs most frequently within the first 5 years after diagnosis in estrogen receptor (ER)-negative breast cancer, those with ER-positive tumors remain at higher risk for recurrence at later times, including decades later. Tamoxifen use greatly reduces the 5-year recurrence risk of ER-positive tumors, but the annual increase in risk of recurrence is still 2% for at least 15 years [2]. Remarkably, prolonged tamoxifen use seems to further reduce the onset of metastasis [3]. Once distant recurrence occurs, patients usually have a poor prognosis [1].

Original lymph node (LN) presence and tumor diameter are essential clinical variables linked to late (5 to 20 years after diagnosis) recurrence in ER-positive tumors. Tumor grade, Ki-67 positivity and progesterone receptor (PR) status have also been found correlated to recurrence, but only in the first 5 years after diagnosis. Importantly, a considerable risk of late recurrence is still present even among women with T1N0 disease [4]. These clinical risk factors were also identified in another study [5]. In addition, positive surgical margins but no margin widths are associated with higher 5- and 9-year local recurrence risks. These risks remain higher if patients are young, have >4 positive lymph nodes or are subjected to re-excision [6]. Lastly, Black women are also more likely to have higher recurrence scores than non-Hispanic White women [7]. 

To date, the factors that indicate a higher breast cancer recurrence risk in the long term have not been fully characterized. Given the need for more data to help prevent breast cancer recurrence, and so as to guide our clinical follow-up and approaches in these patients, artificial intelligence is being implemented. We previously validated a method that used Bayesian networks and information theory to identify key risk factors for breast cancer metastasis more accurately than other known Bayesian-network learning algorithms [8]. This algorithm, named the Markov Blanket and Interactive Risk Factor Learner (MBIL), learns single and interactive risk factors that have a direct influence on a patient’s outcome. Risk factors that are dependent on other variables to have a causative effect are called interactions [9,10]. In the present study, we applied MBIL through 31 variables to learn both a set of direct risk factors and a set of interactive risk factors for 5-, 10- and 15-year recurrence. This algorithm extracted risk factors from the Lynn Sage Database (LSDS) at Northwestern Memorial Hospital, as previously described [8,11].

## 2. Materials and Methods

### 2.1. Materials

The Lynn Sage Dataset (LSDS) is a de-identified and publicly available clinical dataset about breast cancer that was developed via previous studies [8,11]. It was curated using clinical data from the Lynn Sage Database (LSDB) hosted at the Lynn Sage Comprehensive Breast Center at Northwestern Memorial Hospital and the EHR data hosted at the Northwestern Medicine Enterprise Data Warehouse (NMEDW) of the Northwestern University Feinberg School of Medicine and Northwestern Memorial HealthCare. The LSDS consists of records on 6726 breast cancer patients, which span the period from 2 March 1990 to 28 July 2015 [11]. The datapoints queried were included in the clinical records of these patients and were extracted as reported clinically. LSDS for Metastasis (LSM) datasets were retrieved from the LSDS, which focus on breast cancer metastasis as a binary outcome [8]. Not including the binary outcome variable, these datasets contain 31 variables, defined in the following table (Table 1).

### 2.2. Methods

Bayesian networks (BNs) have become a leading architecture for modeling uncertain reasoning in artificial intelligence and machine learning. A Medline search reveals that 3910 papers contained the term “Bayesian network” from 2003 to 2017, while only 252 contained that term from 1993 to 2002. A Bayesian network consists of a directed acyclic graph (DAG), the node set of which contains random variables, and the conditional probability distribution of every variable in the network given each set of values of its parents [12,13].

In general, the Markov blanket of a node T in a Bayesian-network model consists of all parents of T, children of T and parents of children of T [14]. Figure 1 [8] shows a Bayesian-network DAG structure in which the node T is a leaf node because it has no children. So, the Markov blanket of T only consists of its parents, namely nodes X11 through X15. If we run a machine-learning algorithm without knowing the BN DAG structure, nodes X1 through X10, X16 and X17 would all be learned as risk factors of T because these nodes can pass information to T through the parent nodes; i.e., the direct risk factors of T, even though they do not have a direct influence on T. Hence, when learning a BN DAG, we can identify the direct risk factors of a node T via a Markov blanket. This helps to get rid of the background noise which often affects the prediction performance. By incorporating our previous work concerning learning interaction from data [9,10], we developed the Markov Blanket and Interactive Risk Factor Learner (MBIL) method, which can not only identify the Markov blanket of nodes like T but also detect interactive risk factors of nod like T [8]. We evaluated the MBIL using 240 simulated datasets and compared it with three existing detection algorithms developed by other researchers [8]. Interactive risk factors work together to have a nonadditive joint-effect on target nodes such as T. In Figure 1, there are two groups of interactive risk factors of T, nodes X13 and X14 and nodes X8 and X9. The MBIL detects all direct risk factors included in the Markov blanket of target nodes like T, both single ones, like nodes X11, X12 or X15, and interactive ones, like nodes X13 and X14. The MBIL also detects all other interactive risk factors, such as nodes X8 and X9, regardless of whether they are included in the Markov blanket.

In this research we applied the MBIL to learn the direct and interactive risk factors of 5-, 10- and 15-year breast cancer metastases. The MBIL takes a score-based structure-learning approach to learn the Markov blanket of a node. The Bayesian score is the probability of the data given the BN DAG [15]. It measures how well a BN DAG represents the data. We used the Bayesian Dirichlet equivalent uniform (BDeu) score [16] as our score criterion, which is a variant of the Bayesian score. Ideally, we would like to learn a model that represents the reality perfectly. However, due to various reasons, such as the complexity of real-world problems and the limitations of data collected, it is often impossible to learn such a perfect model. Instead, a major task of machine learning is to adjust parameters in order to learn the models that represent the data most closely. With the MBIL we can adjust alpha, also called the Prior Equivalent Sample Size (PESS), which is a parameter built into the BDeu score [16]. Adjusting alpha can affect the complexity of the BN models that are learned from data. This somewhat resembles the fishing activity, in which we adjust the size of fishnet holes to govern the sizes of the seafood we catch. In this study, we ran the MBIL using three different values of alpha, namely alpha = 1, 120 and 480, and each respectively when learning risk factors for 5-year, 10-year and 15-year metastases. The three values of alpha were chosen heuristically, since there are no specific rules as to how alpha values should be chosen. In our previous research, alpha = 240 was used, with which the MBIL identified risk factors that were previously reported in the literature [8]. In this research, we focused on exploring alpha values that were bigger (480) or smaller (1 and 120) than 240 to see how these different alpha values could affect the “discoveries”. The MBIL reported both a set of interactive risk factors and a set of direct risk factors for each of the alpha values. Note that the set of direct risk factors learned by the MBIL included both single and interactive ones.

## 3. Results

Using the MBIL algorithm, we previously described the superior effectiveness of this method in learning direct risk factors that were also present in the literature in the context of 5-year breast cancer metastasis [8]. In the present study, we first used the MBIL to search causal sets of 5-, 10- and 15-year breast cancer metastases. The MBIL produced an output with three alpha values, 1, 120 and 480, and a list of interacting risk factors for each alpha value and time to metastasis. These learned interactions were ranged by their Bayesian scores from the highest to lowest, which was the probability of the data given the Bayesian-network model [8]. Additionally, an output was originated with direct causal sets of metastases. These were a few risk factors that necessarily interacted together to have an effect on the studied outcome [8].

We found that some of the variables identified by the MBIL, such as HER2, were frequently associated with low alpha values for all 5-, 10- and 15-year outcomes. As alpha became stronger, the strength of HER2 as a risk factor of metastatic breast cancer (mBC) recurrence declined, while ER became a stronger risk factor, meaning that while HER2 was identified by the MBIL as an independent causative of mBC, stronger interactions between ER and other variables were necessary in order to identify risk of occurrence of mBC. 

### 3.1. Causal Sets of Metastases

When observing causal sets directly related to metastatic breast cancer, the MBIL found that, at 5 years, direct causal sets of metastases with alpha of 1 were lymph-node positivity and the interaction of triple-negative breast cancer (TNEG) with HER2 assessments (Figure 2); with a more stringent alpha of 120, a direct causal set at 5 years was the interaction of ER, n-TNM and surgical margins; and with the highly interacting alpha of 480, the causal sets were stage, TNEG and ER in interaction with n-TNM and surgical margins (Figure 2). 

At 10 years, disease stage was a sole causal set of metastases with an alpha of 1; with an alpha of 120, having MRI evaluations within 60 days of surgery was a direct causal risk factor for metastases, and ER, n-TNM and surgical margins interacted to form direct causal sets. The 15-year-metastasis causal sets were stage, MRIs_60_surgery and the interaction of ER, n-TNM and surgical margins. With an alpha of 480, 5-year causal sets were the age at diagnosis, menopausal status and lymph node status; at 10 years the causal sets were invasiveness of the tumor and the interaction of age at diagnosis, menopausal status and lymph node status (Figure 2). 

To determine risk of mBC after 15 years, the causal sets were the interaction of lymph node status, histology and invasive tumor location and the interaction of n-TNM, histology and invasive tumor location (Figure 2).

### 3.2. Learned Interactions

Using the MBIL algorithm, we searched for direct interactions that were risk factors of metastatic breast cancer. We first investigated the absolute frequency of known variables impacting breast cancer prognosis. Mainly, we calculated the frequency of ER, HER2, TNEG and tumor grade, as these were the strongest biological determinants of the more aggressive disease present in our dataset [17]. We found that the influence of ER as a risk factor was most frequently found when looking at 5-year metastases with an alpha of 120 (Figure 3). The presence of HER2 among interactive risk factors was highest with an alpha of 1, with decreasing abundance as the time to late relapse became longer, suggesting that HER2 scored higher as an independent structure likely to increase the risk of 5-year metastasis rather than that of 10- or 15-year metastases. TNEG was strongest as a risk factor for tumor metastasis at 15 years when observing interactions using an alpha of 1, supporting its strength as a likely independent risk factor of metastasis [18]. It was present to influence 5-year metastasis at an alpha of 1 when in conjunction with HER2. At an alpha of 120, TNEG was a risk factor of 5-year metastasis when in conjunction with smoking and n-TNM in one interaction and with n-TNM and invasive_tumor_location in another interaction. In addition, TNEG was found to correlate with 10- and 15-year metastases in one interaction; for both risk factors, TNEG interacted with n-TNM and surgical_margins. Finally, at an alpha of 480, TNEG was only found to correlate with 10-year metastasis in one interaction with n-TNM and surgical margins.

We next found the tumor grade to be a strong risk factor of 5-year metastasis with an alpha of 1, whereas with higher alpha values its frequency in interactions predicting metastasis was reduced. Hence, tumor grade appeared to be a strong independent risk factor of 5-year metastasis.

We then calculated the absolute frequency of smoking/alcohol and race/ethnicity in order to strengthen the available evidence regarding these risk factors for late recurrence [19]. We found that smoking and/or alcohol were only present in one interaction for 15-year metastases with an alpha of 1. However, they were present in one interaction for 5- and 15-year metastases with an alpha of 120, and in up to four interactions influencing 5-year metastasis at an alpha of 480 (Figure 3). The presence of the risk factors race and ethnicity was higher as time increased from 5- to 15-year recurrence, particularly at an alpha of 480, suggesting that race might act as a dependent variable to favor late recurrence (Figure 3).

We next calculated the frequency with which these variables interacted with any other variable to constitute direct risk factors of metastases (Table 1, Table 2 and Table 3). At an alpha of 1, ER interacted with only n-TNM, HER2 and LN positive 33% of the time with each, suggesting that the presence of ER may only be a risk factor for metastasis when it is present in conjunction with these three variables (Table 2). ER was found to have more interactions with an alpha of 120, but most frequently with n-TNM (33%), surgical margins (13%) and lymph node positive (13%) (Table 3). The variables that interacted most frequently with ER at an alpha of 480 were n-TNM (32%), surgical margins (16%) and race (16%) (Table 4).

With an alpha of 1, TNEG was found to interact with LN positive/status (29%), HER2 (14%), age at diagnosis (14%), ethnicity (14%), stage (14%) and re_excision (14%). The most frequent interaction was LN positive (28%), which related to 15-year recurrences (Table 2). With an alpha of 120, the most interactive variable of TNEG was n-TNM (50%), followed by surgical margins (25%), smoking (12.5%) and invasive tumor location (12.5%) (Table 3). With an alpha of 480, TNEG was found to only interact with n-TNM (50%) and surgical margins (50%) to influence 10-year recurrence (Table 4).

With an alpha of 1, HER2 was found to interact most frequently with stage (26%), MRIs_60_surgery (43%), ER percent (43%) and TNEG (43%) (Table 2). The strength of HER2 as a risk factor for late recurrence was reduced with alpha values of 120 and 480. In this regard, with an alpha of 120, HER2 interacted most frequently with stage (42%) and surgical margins (17%) (Table 3). With an alpha of 480, HER2 only interacted with surgical margins (57%), stage (29%) and t-TNM (14%) (Table 4).

Race and ethnicity were counted as a single variable for the purposes of quantitation. At an alpha of 1, race and ethnicity interacted once with histology (12.5%), grade (12.5%), ER_percent (12.5%), n-TNM (12.5%), side (12.5%), LN positive/status (12.5%), TNEG (12.5%) and stage (12.5%) (Table 2). With an alpha of 120, interactions of race/ethnicity were more frequent with ER (18%), n-TNM (18%), stage (18%) and LN positive/status (18%) (Table 3). At an alpha of 480, race and ethnicity interacted more frequently with stage (27%), ER (17%) and n-TNM (17%) (Table 4).

Lastly, smoking and alcohol were also combined for the purposes of quantitation, as they followed the same trend of occurrence in the analyses. These appeared to interact, at an alpha of 1, with LN positive/status to act as risk factors of 15-year metastasis (Table 2). At an alpha of 120, smoking and alcohol interacted with TNEG (25%), n-TNM (25%), stage (25%) and histology (25%) (Table 3). Finally, at an alpha of 480, smoking and alcohol were found to interact most frequently with n-TNM (25%) and stage (25%), followed by t-TNM (16.7%), surgical margins (16.7%), ER (8.3%) and race (8.3%) (Table 4).

## 4. Discussion

Advances in the treatment of breast cancer after surgical resection of the primary lesion have altered our approach to those women who are at risk for recurrences. As we now can see metastatic recurrence in the two to three decades after the primary lesion has been removed [20], there is a need to personalize follow-up care based on the likelihood of finding cancer recurrence. For this reason, much effort has gone into determining which clinical-pathological features can predict longer-term outcomes. Classical methods have found a number of characteristics that increase the recurrence risks, but mainly these have been studied as singular parameters correlated to early recurrence, usually in the setting of HER2-positivity or later recurrence in the HER2-negative and ER+ settings [21]. To broaden the coverage provided by molecular and pathological predictors of therapy response, including prior machine-learning approaches [22], we used Bayesian-network machine learning to find not just independent factors for later recurrences but also sets of risk factors that in aggregate could also help in decision making. 

The MBIL algorithm escalates the degree of interactivity between parameters. With the alpha set at the bottom level of 1, known correlates of recurrence were found, including TNEG, HER2 positivity and TNM stage. However, as the alpha was elevated to define interacting sets of parameters, race, age, alcohol and smoking were scored as parts of the prognostic sets. Interestingly, alcohol and smoking were more often linked to recurrences at 5 years than at 15 years; this suggests that pathobiological effects are either short term or reversible at the scale of years to a decade [23,24]. Race and ethnicity have also been implicated in higher recurrence risks, as shown by the 21-gene recurrence score [7]. In our study, race and ethnicity were more often linked to later recurrences, providing for partial personalization of follow-up as these are non-modifiable parameters.

These findings need to be validated in additional cancer databases and with other machine-learning methods. This is particularly true of some confusing denotations. For instance, TNEG was related to later recurrences, which goes against the well-documented clinical course of TNBC usually recurring within three years [21]—and if not, by five years—after which the disease is considered cured if no recurrences are noted. However, the association found herein may simply reflect a statistical anomaly indicating that absence of TNEG means less likelihood of rapid recurrence and therefore any recurrence that happens is more likely to occur after 10 or 15 years. This and other prognostic situations need to be refined in further studies. Still, this work does point to the value of these machine-learning algorithms in discerning prognostic sets at a level of resolution (that of years after primary cancer diagnosis) in addition to the classical methods of biomarker development.

## 5. Conclusions

The MBIL may guide the identification of direct causal sets and interactive risk factors of late breast cancer recurrence. Application of this and similar machine-learning methods is encouraged in further databases to help interpret risk of late mBC.

## Figures and Tables

**Figure 1 cancers-14-00253-f001:**
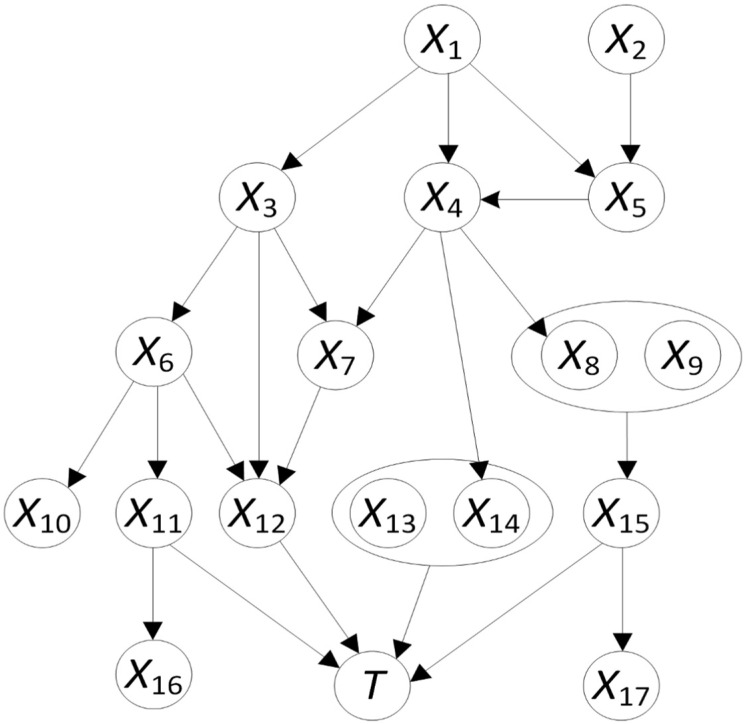
A BN DAG model illustrating the Markov blanket. The Markov blanket of T consists of nodes X11, X12, X13, X14 and X15. These nodes are the direct risk factors of T and separate T from the influence of the noisy variables X1–X10, X16 and X17 (adapted from [8]).

**Figure 2 cancers-14-00253-f002:**
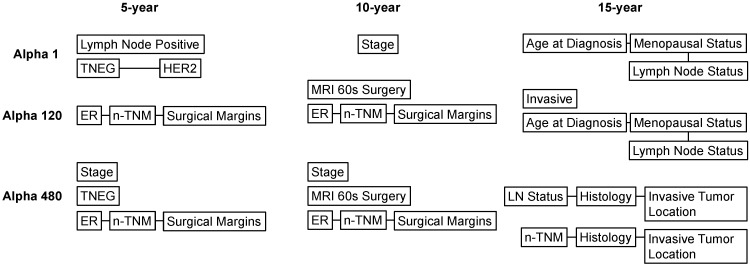
MBIL-generated causal sets of 5-, 10- and 15-year breast cancer metastases.

**Figure 3 cancers-14-00253-f003:**
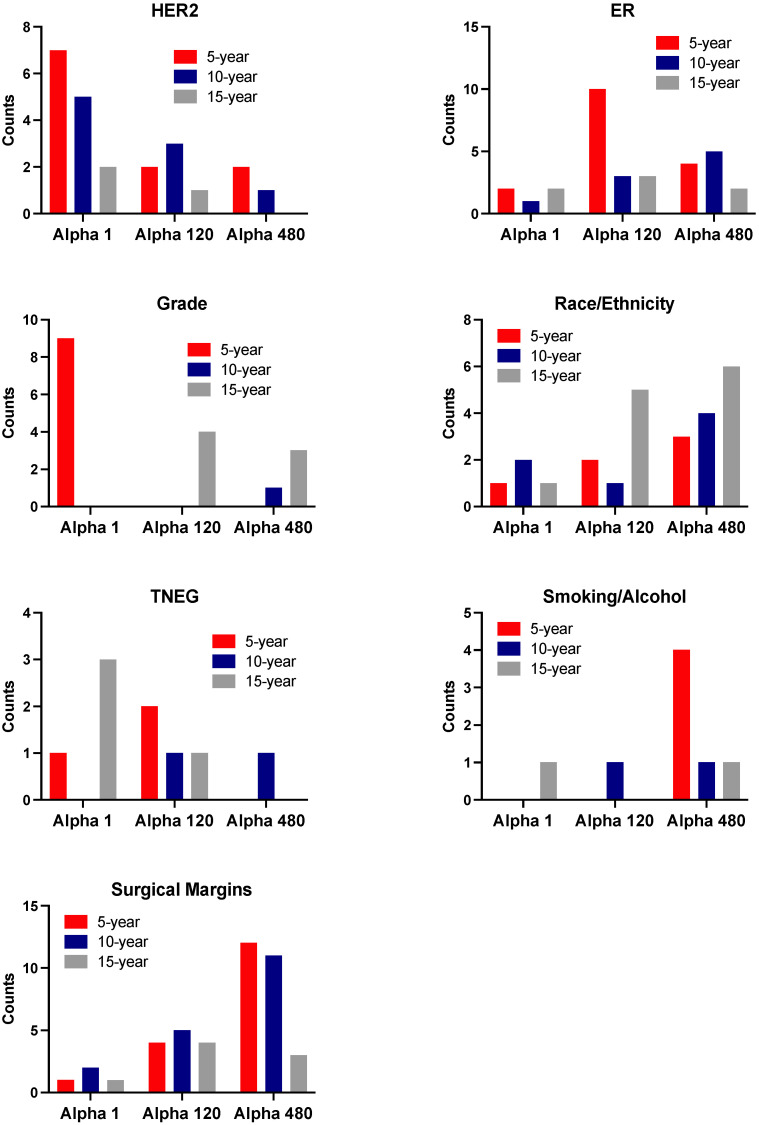
The MBIL generated an output of clinical interactions related to 5-, 10- and 15-year breast cancer metastases. HER2, ER, grade, race/ethnicity, TNEG, smoking/alcohol and surgical margins are represented. Each bar plot indicates the number of counts in which each of these variables was identified as a risk factor of metastasis at different values of alpha.

**Table 1 cancers-14-00253-t001:** Definitions of variables in LSM datasets.

Variables Included	Description	Values
Race	Race of patient	White, Black, Asian, American Indian or Alaskan native, native Hawaiian or other Pacific islander
Ethnicity	Ethnicity of patient	Not Hispanic, Hispanic
Smoking	Smoking history of patient	Ex-smoker, non-smoker, cigarettes, chewing tobacco, cigar
Alcohol usage	Alcohol usage of patient	Moderate, no use, use but not otherwise specified former user, heavy user
Family history	Family history of cancer	Cancer, no cancer, breast cancer, other cancer, cancer but not otherwise specified
Age_at_diagnosis	Age at diagnosis of the disease	0–49, 50–69, >69
Menopausal_status	Inferred menopausal status	Pre-, post-
Side	Side of tumor	Left, right
TNEG	Triple negative status in terms of patient being ER-, PR- and HER2-negative	Yes, no
ER	Estrogen receptor expression	Neg, pos, low pos
ER_percent	Percent of cell stain pos for ER receptors	0–20, 20–90, 90–100
PR	Progesterone receptor expression	Neg, pos, low pos
PR_percent	Percent of cell stain pos for PR receptors	0–20, 20–90, 90–100
P53	Whether P53 is mutated	Neg, pos, low pos
HER2	HER2 expression	Neg, pos
t_tnm_stage	Prime tumor stage in TNM system	0, 1, 2, 3, 4, IS, 1 mic, X
n_tnm_stage	Number of nearby cancerous lymph nodes	0, 1, 2, 3, 4, X
Stage	Composite of size and number of positive nodes	0, 1, 2, 3
Lymph_nodes_removed	Number of lymph nodes removed	0–11, 12–22, >22
Lymph_nodes_positive	Number of positive lymph nodes	0, 1–8, >8
Lymph_node_status	Whether patient has any positive lymph nodes	Neg, pos
Histology	Tumor histology	Lobular, ductal
Size	Size of tumor in mm	0–32, 32–70, >70
Grade	Grade of disease	1, 2, 3
Invasive	Whether tumor is invasive	Yes, no
Histology2	Tumor histology subtypes	IDC, DCIS, ILC, NC
Invasive_tumor_location	Where invasive tumor is located	Mixed duct and lobular, duct, lobular, none
DCIS_level	Type of ductal carcinoma in situ	Solid, apocrine, cribriform, dcis, comedo, papillary, micropapillary
Re_excision	Removal of an additional margin of tissue	Yes, no
Surgical_margins	Whether there are any residual tumors	Residual tumor, no residual tumor, no primary site surgery
MRIs_60_surgery	MRIs within 60 days of surgery	Yes, no

**Table 2 cancers-14-00253-t002:** Variables interacting with an alpha of 1. ER, TNEG, HER2, race/ethnicity and alcohol/smoking are represented with their interacting variables, the number of times they interacted, the years after diagnosis when these interactions were risk factors for metastases, the total number of times the variables interacted and frequency of interaction.

Alpha 1					
Variable	Interacts with	n Times	Years after DG	Total	%
ER	n-TNM	2	5, 10	6	33.33%
ER	HER2	2	5, 15	6	33.33%
ER	LN positive	2	15, 15	6	33.33%
TNEG	HER2	1	5	7	14.29%
TNEG	Age at DG	1	15	7	14.29%
TNEG	LN positive/status	2	15	7	28.57%
TNEG	Ethnicity	1	15	7	14.29%
TNEG	Stage	1	15	7	14.29%
TNEG	Re_excision	1	15	7	14.29%
HER2	Stage	6	5, 5, 10, 10, 10, 15	23	26.09%
HER2	MRIs_60_surgery	1	5	23	4.35%
HER2	ER_percent	1	5	23	4.35%
HER2	TNEG	1	5	23	4.35%
HER2	Histology	3	5, 10, 10	23	13.04%
HER2	Grade	2	5, 5	23	8.70%
HER2	Invasive tumor location	2	5, 10	23	8.70%
HER2	ER	1	5, 15	23	4.35%
HER2	PR	2	5, 10	23	8.70%
HER2	LN positive/status	3	10, 10, 15	23	13.04%
HER2	Surgical margins	1	10	23	4.35%
Race/ethnicity	Histology	1	5	8	12.50%
Race/ethnicity	Grade	1	5	8	12.50%
Race/ethnicity	ER_percent	1	10	8	12.50%
Race/ethnicity	n-TNM	1	10	8	12.50%
Race/ethnicity	Side	1	10	8	12.50%
Race/ethnicity	LN positive/status	1	10	8	12.50%
Race/ethnicity	TNEG	1	15	8	12.50%
Race/ethnicity	Stage	1	15	8	12.50%
Alcohol/smoking	LN positive/status	1	15	1	100.00%

**Table 3 cancers-14-00253-t003:** Variables interacting with an alpha of 120. ER, TNEG, HER2, race/ethnicity and alcohol/smoking are represented with their interacting variables, the number of times they interacted, the years after diagnosis when these interactions were found to be risk factors for metastases, the total number of times the variables interacted and frequency of interaction.

Alpha 120					
Variable	Interacts with	n Times	Years after DG	Total	%
ER	n-TNM	9	5, 5, 5, 5, 5, 5, 5, 10, 15	30	30.00%
ER	Surgical margins	4	5, 5, 10, 15	30	13.33%
ER	Family history	2	5, 10	30	6.67%
ER	LN positive/status	4	5, 10, 15, 15	30	13.33%
ER	HER2	1	5	30	3.33%
ER	MRIs_60_surgery	1	5	30	3.33%
ER	Race/ethnicity	3	5, 15, 15	30	10.00%
ER	Histology	1	5	30	3.33%
ER	Invasive tumor location	1	5	30	3.33%
ER	Size	1	5	30	3.33%
ER	Side	1	5	30	3.33%
ER	DCIS_level	2	5, 10	30	6.67%
TNEG	n-TNM	4	5, 5, 10, 15	8	50.00%
TNEG	Surgical margins	2	10, 15	8	25.00%
TNEG	Smoking	1	5	8	12.50%
TNEG	Invasive tumor location	1	5	8	12.50%
HER2	ER	1	5	12	8.33%
HER2	n-TNM	1	5	12	8.33%
HER2	Stage	5	5, 10, 10, 10, 15	12	41.67%
HER2	Surgical margins	2	5, 10	12	16.67%
HER2	Histology	1	10	12	8.33%
HER2	Grade	1	15	12	8.33%
HER2	PR	1	10	12	8.33%
Race/ethnicity	ER	3	5, 15, 15	17	17.65%
Race/ethnicity	n-TNM	3	5, 10, 15	17	17.65%
Race/ethnicity	Stage	3	5, 15, 15	17	17.65%
Race/ethnicity	Surgical margins	1	5	17	5.88%
Race/ethnicity	ER_percent	1	10	17	5.88%
Race/ethnicity	Grade	1	15	17	5.88%
Race/ethnicity	LN positive/status	3	15	17	17.65%
Race/ethnicity	Re_excision	1	15	17	5.88%
Race/ethnicity	PR_percent	1	15	17	5.88%
Smoking/alcohol	TNEG	1	5	4	25.00%
Smoking/alcohol	n-TNM	1	5	4	25.00%
Smoking/alcohol	Stage	1	10	4	25.00%
Smoking/alcohol	Histology	1	10	4	25.00%

**Table 4 cancers-14-00253-t004:** Variables interacting with an alpha of 480. ER, TNEG, HER2, race/ethnicity and alcohol/smoking are represented with their interacting variables, the number of times they interacted, the years after diagnosis when these interactions were found to be risk factors for metastases, the total number of times the variables interacted and frequency of interaction.

Alpha 480					
Variable	Interacts with	n Times	Years after DG	Total	%
ER	n-TNM	6	5, 5, 5, 10, 10, 10	19	31.58%
ER	Surgical margins	3	5, 5, 10	19	15.79%
ER	Race	3	5, 10, 15	19	15.79%
ER	Size	1	5	19	5.26%
ER	Smoking	1	5	19	5.26%
ER	Family history	1	10	19	5.26%
ER	LN positive/status	1	10	19	5.26%
ER	Stage	1	10	19	5.26%
ER	DCIS_level	1	10	19	5.26%
ER	Age at DG	1	10	19	5.26%
TNEG	n-TNM	1	10	2	50.00%
TNEG	Surgical margins	1	10	2	50.00%
HER2	Stage	2	5, 10	7	28.57%
HER2	Surgical margins	4	5, 5, 10	7	57.14%
HER2	t-TNM	1	5	7	14.29%
Race/ethnicity	Stage	6	5, 15, 15, 15, 15, 15	22	27.27%
Race/ethnicity	Surgical margins	2	5, 5	22	9.09%
Race/ethnicity	ER	3	5, 10, 15	22	13.64%
Race/ethnicity	n-TNM	3	5, 10, 10	22	13.64%
Race/ethnicity	Family history	1	10	22	4.55%
Race/ethnicity	LN positive/status	1	10	22	4.55%
Race/ethnicity	ER_percent	1	10	22	4.55%
Race/ethnicity	Grade	1	15	22	4.55%
Race/ethnicity	Invasive tumor location	1	15	22	4.55%
Race/ethnicity	Re-excision	1	15	22	4.55%
Race/ethnicity	Alcohol	1	15	22	4.55%
Race/ethnicity	Histology2	1	15	22	4.55%
Smoking/alcohol	t-TNM	2	5, 5	12	16.67%
Smoking/alcohol	n-TNM	3	5, 5, 5	12	25.00%
Smoking/alcohol	Stage	3	5, 10, 15	12	25.00%
Smoking/alcohol	Surgical margins	2	5, 10	12	16.67%
Smoking/alcohol	ER	1	5	12	8.33%
Smoking/alcohol	race	1	15	12	8.33%

## Data Availability

Data available in a publicly accessible repository that does not issue DOIs. Publicly available datasets were analyzed in this study. This data can be found here: datadryad.org (accessed on 29 November 2021) [11].

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
