# Peer review of "Machine Learning to Discern Interactive Clusters of Risk Factors for Late Recurrence of Metastatic Breast Cancer"

_cancers, 2022, doi:10.3390/cancers14010253_

Round 1

Reviewer 1 Report

This paper examines machine-learning to retrieve single and interactive 16 clinical and pathological risk factors of 5-, 10- and 15-year metastasis of breast cancer. The current study uses a previously validated algorithm named the Markov Blanket and 25 Interactive risk factor Learner (MBIL). This manuscript is well written.  Here are some suggestions for improvement:

  • MBIL uses three different values of alpha, namely, alpha =1, 120, and 480 each respectively when evaluating risk factors for 5-year, 10-year, or 15-year metastasis. It is unclear how the three values of alpha determined. It is also unclear if any goodness of fit tests were performed to determine the selection process.
  • It is unclear how smoking, alcohol and race/ethnicity were defined, and what rationale was used to combine some of the variables. A more detailed description of the methods is needed.
  • Understanding risk factors for recurrence is very important. Please list the risk factors that were evaluated briefly. The authors have mentioned their previous published work, but it would be helpful to the readers to understand the risk factors considered without referring to the original paper.
  • HER2, ER, Grade, Race/ethnicity, TNEG, Smoking/Alcohol, and Surgical Margins were considered in MBIL clinical interactions predictive of 5-, 10- and 15-year breast cancer metastasis. Was PR status considered? In addition to smoking and alcohol, were BMI and physical activity considered?

Author Response

Reviewer 1:

This paper examines machine-learning to retrieve single and interactive 16 clinical and pathological risk factors of 5-, 10- and 15-year metastasis of breast cancer. The current study uses a previously validated algorithm named the Markov Blanket and 25 Interactive risk factor Learner (MBIL). This manuscript is well written.  Here are some suggestions for improvement:

  • MBIL uses three different values of alpha, namely, alpha =1, 120, and 480 each respectively when evaluating risk factors for 5-year, 10-year, or 15-year metastasis. It is unclear how the three values of alpha determined. It is also unclear if any goodness of fit tests were performed to determine the selection process.

Response: The three values of alpha were chosen heuristically. As described in the Methods (section 2.2), alpha is a parameter that was introduced in a classic Bayesian score called BDeu. There are no specific rules as to how alpha values should be chosen. In our previous research, alpha=240 was used, at which MBIL identified an interactive risk factor that were supported by literature. In this research, we explored alpha values that are larger (480) or smaller (1 and 120) than 240 and hoped to see how these different alpha values can affect the “discoveries”. This is now made clear in the Methods section.

            Since we were not able to make any assumptions in terms of the “expected values/distribution” of alpha, we were not able to conduct goodness of fit tests during the selection process of alpha values.

  • It is unclear how smoking, alcohol and race/ethnicity were defined, and what rationale was used to combine some of the variables. A more detailed description of the methods is needed.

Response: These datapoints were extracted from the clinical record that was mined as part of building the LSDS. This is now included in the new section of Materials (section 2.1). Race/ethnicity and smoking/alcohol were combined since when they were separate there was not a good way to quantify, but individually they seemed to both follow the same trend of occurrence in the analysis.

  • Understanding risk factors for recurrence is very important. Please list the risk factors that were evaluated briefly. The authors have mentioned their previous published work, but it would be helpful to the readers to understand the risk factors considered without referring to the original paper.

Response: These are now listed in the new table 1.

.

  • HER2, ER, Grade, Race/ethnicity, TNEG, Smoking/Alcohol, and Surgical Margins were considered in MBIL clinical interactions predictive of 5-, 10- and 15-year breast cancer metastasis. Was PR status considered? In addition to smoking and alcohol, were BMI and physical activity considered?

Response: The data variables in the LSM (LSDS for metastasis) are now described in the Materials sections and the new table 1.

Reviewer 2 Report

Further evaluations should be carried out and the results compared with the other ML algorithm.

As the authors claim that they used Bayesian network algorithms for prediction, there are no clear outcomes of this, and of course no comparison with other techniques.

Author Response

Reviewer 2:

Further evaluations should be carried out and the results compared with the other ML algorithm.

As the authors claim that they used Bayesian network algorithms for prediction, there are no clear outcomes of this, and of course no comparison with other techniques.

Response: This report is not about prediction, rather it is about applying MBIL to “detect/discover” new risk factors from real datasets; the paper is now rewritten to emphasize this point. MBIL is not a prediction algorithm rather it is a detection algorithm. This type of algorithm is normally evaluated using simulate datasets due to lack of “gold standard” when applied to real datasets. We evaluated MBIL using 240 simulated datasets and also compared MBIL with three existing detection algorithms developed by other researchers. These results were described in our “methodology” paper concerning MBIL, which was previously published and referred in this “application” paper. We have now included a description of the evaluation work we did with MBIL in the “Methods” section (section 2.2).

Reviewer 3 Report

The manuscript describes well the methods used. However, it is too technical and very difficult to undertand. The manuscript should be extensively modified in order to render it undestandable to a huge audience. 

In addition, it is not evident what kind of risk factors have been analyzed and should be well specified.

In introduction section there are no description of the different types of breast cancers, as triple negative ones. HER2 was analyzed in results section, but in introduction its role is not described. Androgen receptor displays also a role in breast cancer, but it is not mentioned.

in results section should be indicated the meaning of alfpa1 and etc.

what is the meaning of "the interaction of TNEG with HER2 assessments? HER expression? HER mutation?

in paragraph 3.2 what is the meaning of "ER was most frequent"?

Authors anayzed ER alpha? A mention to ER beta should be done.

There are several therapeutic options that should be mentioned. 

Author Response

Reviewer 3:

The manuscript describes well the methods used. However, it is too technical and very difficult to undertand. The manuscript should be extensively modified in order to render it understandable to a huge audience. 

In addition, it is not evident what kind of risk factors have been analyzed and should be well specified.

Response: These are now listed in the new table 1.

In introduction section there are no description of the different types of breast cancers, as triple negative ones. HER2 was analyzed in results section, but in introduction its role is not described. Androgen receptor displays also a role in breast cancer, but it is not mentioned.

Response: All types of breast cancer were included in the LSDS and are queried here without separating the subtypes. For this reason there are linkages to HER2, TNEG and ER status. Androgen receptor was not routinely examined in these breast cancers and thus not included.

in results section should be indicated the meaning of alfpa1 and etc.

Response: This is now included in the Methods (section 2.2).

what is the meaning of "the interaction of TNEG with HER2 assessments? HER expression? HER mutation?

Response: These are now explained in the new table 1.

in paragraph 3.2 what is the meaning of "ER was most frequent"?

Response: This is now clarified in the manuscript.

Authors analyzed ER alpha? A mention to ER beta should be done.

Response: Alpha refers to a parameter in Bayesian algorithm, not to the transcription factor. Our results were not able to provide information regarding detection of the ER alpha or beta isoforms and their role as risk factors.

There are several therapeutic options that should be mentioned. 

Response: As this study just detects risk factors, we do not want to imply direct therapeutic options at present, so as not to imply linkages stronger than they might be.

Round 2

Reviewer 1 Report

The revised manuscript is responsive to the reviewers' comments. I have no further comments.

Reviewer 3 Report

Authors addressed exaustively the concerns. I have no further comments.